# Plasma microRNA signatures predict prognosis in canine osteosarcoma patients

Latasha Ludwig[1☯¤a], Michael Edson[1☯¤b], Heather Treleaven[1], Alicia M. Viloria-Petit[2], Anthony J. Mutsaers[2,3], Roger Moorehead[2], Robert A. Foster[1], Ayesha Ali[4], R. Darren Wood[1], Geoffrey A. Wood[1]*

1 Department of Pathobiology, Ontario Veterinary College, University of Guelph, Guelph, Ontario, Canada, 2 Department of Biomedical Sciences, Ontario Veterinary College, University of Guelph, Guelph, Ontario, Canada, 3 Department of Clinical Studies, Ontario Veterinary College, University of Guelph, Guelph, Ontario, Canada, 4 Department of Mathematics and Statistics, University of Guelph, Guelph, Ontario, Canada

☯ These authors contributed equally to this work.
¤a Current address: Department of Population Medicine and Diagnostic Sciences, College of Veterinary Medicine, Cornell University, Ithaca, New York, United States of America
¤b Current address: MaRS Centre, University Health Network, Toronto, Ontario, Canada
* gewood@uoguelph.ca

**Data Availability Statement:** All relevant data are within the manuscript and its Supporting Information files.

## Abstract

Appendicular central osteosarcoma (OSA) is a common and highly aggressive tumour in dogs. Metastatic disease to the lungs is common and even with chemotherapy the prognosis is generally poor. However, few cases survive well beyond reported median survival times. Current methods, including histologic grading schemes, have fallen short in their ability to predict clinical outcome. MicroRNAs (miRNAs) are small molecules present in all tissues and bodily fluids and are dysregulated in cancer. Previous studies have demonstrated the diagnostic and prognostic potential of miRNAs in canine OSA. We sought to investigate multiple miRNA and multiple variable models for diagnosis and prognosis of canine OSA using plasma samples across three populations of dogs from two veterinary biobanks. Fifty-six miRNAs were analyzed by real-time quantitative polymerase chain reaction. MiR-214-3p was the only miRNA with increased expression across all OSA populations compared to controls. Using a decision tree model for diagnosis, miR-214-3p was the first step in this multi-miRNA model. High expression of miR-214-3p alone was also a predictor of shorter overall survival and disease-free interval across all populations. In both multiple miRNA and multiple variable models, miR-214-3p was always the first decision point with high expression consistently predicting a worse prognosis. Additional miRNAs in combination with low expression of miR-214-3p similarly had a worse prognosis demonstrating better outcome prediction using multiple miRNAs compared to using miR-214-3p alone. Multiple variable models only need to use miRNAs to be predictive although clinical parameters such as age, sex, and tumour location were considered. MiR-214-3p is clearly an important prognostic predictor of canine OSA in plasma as supported by previous studies and across our multiple sample populations. Multiple miRNA models provided superior categorization of patients in predicting clinical outcome parameters compared to the single miRNAs.

**Funding:** G.A.W. 053527; Pet Trust; https://pettrust.uoguelph.ca/ The sponsor had no role in the study design, data collection and analysis, decision to publish, or preparation of the manuscript. G.A.W. 401622; Natural Sciences and Engineering Research Council of Canada; https://www.nserc-crsng.gc.ca/index_eng.asp; The sponsor had no role in the study design, data collection and analysis, decision to publish, or preparation of the manuscript. L.L. Vanier Canada Graduate Scholarship; https://vanier.gc.ca/en/home-accueil.html The sponsor had no role in the study design, data collection and analysis, decision to publish, or preparation of the manuscript.

## Introduction

Appendicular central osteosarcoma (OSA) is a highly aggressive neoplasm and the most common primary tumour of bone in dogs [1]. The median age at diagnosis is approximately 7 years old, with large and giant breeds at increased risk [2]. These dogs typically present to their veterinarian with lameness and/or a possible fracture [3]. The current standard of care includes amputation of the affected limb and adjuvant cytotoxic chemotherapy [3, 4]. The median survival time with standard-of-care treatment is approximately 1 year [5]. Although most dogs do not have radiographically detectable metastases at the time of diagnosis, 90–95% have micrometastases that later become detectable and are the main cause of mortality [6, 7]. There is a wide range in the survival time of canine OSA patients with some succumbing to disease within weeks, while a few live for over two years [8]. Although additional clinical trials and novel therapies have been investigated, no significant advances in treatment or survival prediction have emerged in decades [9, 10].

While uncommon, the presence of clinically detectable metastatic disease at diagnosis is a strong predictor of a poorer prognosis, and palliative treatment is often recommended in these cases [3, 7]. Currently, other than checking for metastases, there is no standard system to predict how a patient will respond to treatment or to provide prognostic information to owners and veterinarians to assist in clinical decision-making. Histologic grading schemes are a common and fundamental tool in canine and human pathology [11, 12]. Unfortunately, current grading schemes for OSA were shown to not predict outcome in a large study of canine patients receiving standard-of-care therapy [13]. Serum alkaline phosphatase (ALP) activity is a strong prognostic indicator in some studies but it frequently loses significance in multivariable analyses [13–19]. Several ALP isoforms exist, which can be elevated in many other diseases, and few diagnostic laboratories routinely offer bone-specific ALP testing [1]. Additional biochemical tests and blood parameters such as circulating neutrophils and urokinase plasminogen activator have prognostic utility but require further investigation [16, 17]. The location of the primary tumour has clinical outcome significance, however, which location provides a worse prognosis varies between studies [14, 15]. To provide more reliable prognostic information for canine OSA, additional prognostic factors are required.

A recent review outlined multiple potential novel biomarkers, including microRNAs (miRNAs) [5]. MiRNAs are small non-coding RNA molecules that are present within cells and in bodily fluids including plasma [20]. Their various carrier mechanisms within the blood enhance their stability compared to other RNA species [20]. Several studies demonstrate the potential diagnostic and prognostic ability of miRNAs in plasma and serum in canine appendicular OSA [21–23]. These studies chose patients receiving standard-of-care therapy; they only evaluated a small number of miRNAs (miR-16, miR-126, and miR-214 by Heishima et al., 2019) or a small number of samples from a single population (n = 31 by Dailey et al., 2021). Both studies considered multiple variables in their analyses, including serum ALP activity and tumour location, but the prognostic miRNAs were different between the studies and the results also differed in the importance of other clinical variables.

Evaluation of miRNAs by real-time quantitative polymerase chain reaction (RT-qPCR) in serum and plasma poses two major challenges. First is the appropriate selection of endogenous controls. Traditional RNA molecules used as endogenous controls for larger RNA species are inappropriate when evaluating miRNAs because they do not have a similar stability and thus do not accurately reflect the quality of miRNAs in the sample [24–26]. Tools such as NormFinder and GeNorm are widely used however, the selection criteria for controls are poorly defined in many publications [27]. Secondly, miRNAs are present in erythrocytes [28]. These miRNAs can be released into plasma and serum and so the degree of hemolysis changes the miRNA

profile of the sample [29]. MiR-16 and miR-451a are within erythrocytes and released with hemolysis, however there are many miRNAs for which hemolysis-related expression changes are unknown [28, 30, 31]. Unfortunately, miR-16 is highly expressed, not only in erythrocytes but across many cell types and is historically used as a stable endogenous control [21, 22, 26]. Evaluation of hemolysis in plasma and serum samples is critical. Many studies only use visual inspection, which is an insensitive method [21–23, 30, 31]. Complete exclusion of even slightly hemolyzed samples would make measurement of circulating miRNAs an unrealistic diagnostic test; even in healthy controls obtaining a completely non-hemolyzed sample as determined by sensitive methods such as spectrophotometry, can be challenging.

This study investigates a large profile of miRNAs in plasma as candidates to predict clinical outcome in dogs with appendicular OSA that received standard-of-care treatment at multiple institutions acquired from two different biobanks. Multi-miRNA and multi-variable models were evaluated to determine whether they were better able to predict prognosis over single miRNAs. Patient factors were considered, including weight, tumour location, and serum ALP activity, to determine if combining them with miRNAs provides greater prediction of clinical outcome.

## Materials and methods

### Plasma samples

All owners provided informed consent and an animal utilization protocol was approved by the University of Guelph Animal Care Committee (protocol 4431). Cases were selected based on a confirmed histopathologic diagnosis of appendicular OSA and no clinically detectable meta-static disease that received amputation and at least one dose of cytotoxic chemotherapy. Plasma samples were obtained from the Ontario Veterinary College (OVC) Veterinary Bio-bank (https://icci.uoguelph.ca/ovc-veterinary-biobank/) and the Canine Comparative Oncol-ogy and Genomics Consortium (CCOGC, http://www.ccogc.net). Patients obtained care at the Animal Cancer Center (OVC, Guelph, ON) for samples received from the OVC Veterinary Biobank or from multiple institutions across the United States (Ohio State University, Univer-sity of Wisconsin-Madison, Tufts University, and University of California, Davis) for samples received from the CCOGC. The plasma samples obtained from the OVC Veterinary Biobank were processed across two different time points designated OVC1 (n = 35; amputation between 2015–2019) and OVC2 (n = 13; amputation between 2010–2013), with OVC2 serving as a confirmatory population. The CCOGC population consisted of 13 plasma samples. All samples were collected prior to limb amputation surgery, often within a day, and without prior chemotherapy treatment. Complete follow-up information from diagnosis until death due to a specific cause was discernable from the medical records in 58/61 cases. For the remaining three cases, one from the OVC1 group and two from the CCOGC group, only a last known date alive was available, each with no history of clinical metastatic disease at that time. Plasma samples from healthy control dogs (n = 21) were collected at OVC Health Sciences Center (HSC) from blood donors and staff-owned dogs with informed written consent. All control dogs had a physical examination, as well as a complete blood count and a serum biochemical analysis, to confirm that they were healthy. All blood samples were collected in $K_2$EDTA tubes (BD Vacutainer) and then centrifuged to separate the plasma, which was subsequently stored at -80˚C until analysis.

### MiRNA isolation

MiRNAs were extracted from 200 μL of each plasma sample using the miRNeasy Serum/ Plasma Kit (QIAGEN) and performed at room temperature, unless otherwise specified. For

each sample, 1000 μL of QIAzol lysis reagent was added, vortexed, and left to sit for 5 minutes. Afterwards, 3.5 μL of a *C. elegans* synthetic miRNA (cel-miR-39-3p) and 200 μL of chloroform were added, vortexed, and left to sit for 3 minutes. Samples were centrifuged at 12,000 g for 15 minutes at 4˚C (Sorvall Legend Micro21R Centrifuge; Thermo Scientific). From the top aqueous layer, 600 μL was extracted and added to 900 μL of 100% ethanol. This mixture was added to a RNeasy MinElute spin column and centrifuged at 12,000 rpm (Eppendorf Mini Spin) for 30 seconds. The flow through was discarded and these steps were repeated with 700 μL of RWT buffer, 500 μL of RPE buffer, and 500 μL of 80% ethanol (spun for 2 minutes). The collection tube was replaced, and columns were dried by centrifugation at 14,500 rpm for 5 minutes with the lids open. The collection tube was replaced again and 14 μL of RNase-free water was added to the column before being centrifuged at 14,500 rpm for 1 minute. The extracted RNA was stored at -80˚C until use.

## Target miRNA selection

Similar to Craig et al. (2019), a pilot study of 5 pooled plasma samples for the control and OSA groups was performed utilizing samples from the OVC Veterinary Biobank. The 5 OSA dogs selected were representative of the various breeds, ages, weights, sexes, and primary tumour locations of the population of canine OSA patients enrolled at the OVC HSC. Additionally, 5 control dogs were selected with similar demographics. Each pooled sample used 40 μL of plasma from each of 5 dogs belonging to the same group, to create a final volume of 200 μL for miRNA isolation as above. The pilot study was pursued to identify miRNAs differentially expressed between groups and that also had robust expression in both OSA and control dogs. The commercially available canine miScript miRNome miRNA polymerase chain reaction (PCR) array (QIAGEN) was used to profile 277 miRNAs known to be expressed in dogs (S1 Table). A miScript II RT Kit (QIAGEN) was used to convert extracted miRNA into complementary DNA (cDNA) by reverse-transcription PCR using the manufacturer's protocol and a standard volume of 1.5 μL of extracted RNA. These tubes were then added to a C1000 Thermal Cycler (Bio-Rad) and incubated at 37˚C for an hour followed by inactivation at 95˚C for 5 minutes, then cooled to 4˚C for at least 5 minutes. The lid temperature was set to 95˚C to prevent sample loss by evaporation. This 20 μL cDNA solution was diluted with 90 μL of nuclease-free water and stored at -20˚C until used for RT-qPCR. The corresponding miScript SYBR green kit (QIAGEN) was used per the manufacturer's protocol to create a master mix which was robotically pipetted (10 μL per well) onto the array with an NX$^P$ Automated Workstation (Beckman Coulter). The RT-qPCR was then performed using a Roche LC480 LightCycler set to QIAGEN's recommended settings. Cycle threshold (Ct) values were calculated using the second derivative maximum method, in the LC480 software (release 1.5.1.62 SP3). Fifty-six miRNAs were added to the custom miRCURY PCR array design (QIAGEN) (S2 Table) to be used for the individual samples described above. These were selected based on those with the largest fold-changes between groups based on the above experiments, those of interest for a future study, those reported to be significant in the literature, and QIAGEN's recommended controls. The mature miRNA sequence and cross-species homology for each assay investigated (S2 Table) can be found on GeneGlobe within PCR panel design tool (https://geneglobe.qiagen.com/ca/customize/pcr/mirna/mircury-lna-mirna-custom-pcr-panels) with the respective catalog numbers provided.

## Reverse-transcription PCR

MiRNA isolates were converted into cDNA by reverse-transcription PCR using the miRCURY LNA RT Kit (QIAGEN). PCR tubes were kept on ice and filled with 2 μL of miRCURY RT

Buffer, 5.94 μL of RNase-free water, 1 μL of miRCURY RT Enzyme Mix, 0.5 μL of UniSp6 RNA spike-in, and a standard volume of 0.56 μL of extracted RNA (equivalent to 8 μL input RNA). These tubes were incubated at 42˚C for an hour followed by inactivation at 95˚C for 5 minutes, then cooled to 4˚C for at least 5 minutes (C1000 Thermal Cycler (Bio-Rad)). The lid temperature was set to 95˚C to prevent sample loss by evaporation. Eight μL of this cDNA solution was then diluted with 312 μL of nuclease-free and used immediately for RT-qPCR.

## Real-time quantitative PCR

A master mix was created for each sample using the corresponding miRCURY SYBR green kit (QIAGEN). Each tube received 360 μL of SYBR Green, 288 μL of the diluted cDNA sample, and 72 μL of RNase-free water. This master mix was robotically pipetted to a miRCURY LNA miRNA custom PCR array 384-well plate (QIAGEN) (10 μL per utilized well) with an NX$^P$ Automated Workstation (Beckman Coulter). The RT-qPCR was then performed using a Roche LC480 LightCycler set to a 384-well plate configuration. Volume reaction was set to 10 μL and the detection format was SYBR Green I/HRM Dye. The PCR array was incubated at 95˚C for 2 minutes at a ramp rate of 4.8˚C/s. This was followed by 45 cycles of 10 seconds at 95˚C, ramp rate of 4.8˚C/s, and 60 seconds at 56˚C, ramp rate of 2.5˚C/s. A single acquisition was taken at the end of the 56˚C step in each cycle. Afterwards, a melting curve analysis was performed from 55–95˚C at a ramp rate of 0.11˚C/s with 5 acquisitions taken per second. The PCR array was then cooled to 37˚C. Ct values were calculated using the second derivative maximum method in the LC480 software (release 1.5.1.62 SP3).

## Quality assessment of RT-qPCR results

As per QIAGEN's recommendations, the UniSp3 Inter-plate calibrator (IPC) was utilized to correct for potential differences between PCR arrays. The difference between the average UniSp3 of the no template control (NTC; 6 replicates per plate) for each PCR array and the overall average was determined. Each Ct value was subtracted by this difference to get an IPC-corrected Ct value. All further analyses utilized the IPC-corrected Ct values, from now on simply referred to as Ct values.

A Ct value cut-off of 35.00 was employed for all miRNAs and controls. The amplification and melting curves were manually assessed to ensure appropriate Ct determination by the LC480 software (release 1.15.1.62). All undetectable Ct values, those above 35.00, and those with a visually inappropriate amplification curve were changed to 35.00 to provide a conservative baseline to assess fold-change in expression. All cel-miR-39-3p and UniSp6 values of samples included in the study were within the average of all samples +/- two standard deviations to ensure adequate miRNA isolation and reverse-transcription efficiency. Each sample and miRNA were evaluated for expression levels. All samples included in the study had an overall expression of miRNAs within the average of all samples isolated minus two standard deviations. MiRNAs expressed in fewer samples than the overall miRNA expression average minus two standard deviations were removed from further analysis. This included 6 miRNAs originally included on the custom panels (miR-433-3p, miR-210, miR-589, miR-551a, miR-138b, miR-802).

## Hemolysis assessment

Prior to miRNA isolation, the degree of hemolysis was analyzed in each sample using a Nano-Drop™ 2000 (Thermo Scientific) to determine UV-vis wavelength results for absorbance at 414 nm and 375 nm. All samples had a 414 nm value less than 1.5, or a 414/375 ratio < 1.4 [29, 32, 33]. For each miRNA, hemolysis was assessed in 2 ways. First, the Ct values for each sample

were plotted against their 414 nm absorbance value. The relationship between a hemolysis-affected miRNA and their Ct values is typically linear; therefore, a linear trendline was estimated and an $R^2$ value was calculated [29]. Second, samples were divided into hemolyzed and non-hemolyzed groups and a Mann-Whitney U test was used to test for differences and calculate p-values for each miRNA. Samples were classified as hemolyzed if they had a 414 nm absorbance value $> 0.2$. Fortunato et al. (2014) found lipid content in plasma samples to also affect 414 nm absorbance and used 375 nm absorbance as a normalizer [32]. Therefore, any samples deemed hemolyzed by their 414 nm value which had a 414/375 nm ratio $< 1.4$ were removed from the first analysis (414 nm value to Ct value) and switched to the non-hemolyzed group for the second analysis [32].

## Statistical analysis

All analyses were completed using the R Studio environment, software version 4.1.0. (R Core Team, 2021). First, the NormFinder R script was used to identify stable miRNAs to be used as endogenous controls [34]. MiRNAs selected to undergo NormFinder analysis had to be expressed in all samples (Ct $< 35.00$) and miRNAs that were deemed likely associated with hemolysis, per the above analyses, were removed from consideration. NormFinder recommends between 5–10 genes for optimal function of their algorithm [34]. To satisfy these requirements, 2 additional miRNAs with non-definitive relationships with hemolysis were considered (hsa-let-7c-5p and cfa-miR-140). As recommended by Vandesompele et al. (2002), a minimum of 3 controls were selected and additional miRNAs would only be used if they provided a significant reduction in variability [35]. The top 4 stable miRNAs had similar stability values; therefore, to select endogenous controls the list of stable pairs, as provided by Norm-Finder, were used to select the best combination. Although hsa-let-7c-5p was the second most stable miRNA, it was not selected due to concerns of its relationship with hemolysis based on the above analyses. MiRNAs cfa-miR-23b, rno-miR-223-3p, and hsa-miR-27b-3p were selected for further use as endogenous controls and the addition of a fourth miRNA was found to be unnecessary as it did not provide a significant reduction in variability [35]. Normalization was performed using the comparative Ct method where the Ct for a miRNA of interest was subtracted from the average Ct value of the pre-determined stable endogenous miRNA controls [36]. In place of the arithmetic average, the geometric average was employed as it is less susceptible to outliers [35]. Normalization was completed on all data simultaneously.

Due to an insufficiently large enough sample size for each group, we were unable to assume normality so a Shapiro-Wilk test for normality was used to assess the distributions of each group for each miRNA. If one or more groups failed the normality test (p was $\leq 0.05$), the miRNA was considered non-normally distributed. Forty-five % of miRNAs were found to be non-normally distributed, therefore non-parametric tests were used. Using the normalized Ct values ($\Delta$Ct), the Wilcoxon-rank sum test was performed. To correct for multiple testing, the Benjamini-Hochbreg procedure was used and a p-value of $< 0.05$ was considered statistically significant [37]. This test was repeated for each miRNA comparing each OSA sample population independently to the single control group. Additionally, fold-changes were calculated as $2^{-\Delta\Delta Ct}$, where $\Delta\Delta$Ct is the median $\Delta$Ct for a miRNA of interest in control samples subtracted from the $\Delta$Ct of the miRNA in OSA samples. This was repeated for all OSA samples, and the median was taken for each miRNA. Fold-differences were reported and equivalent to fold-change if $\geq 1$, while fold-differences were equivalent to the negative inverse of the fold-change if $< 1$.

Prior to creating survival curves, the $\Delta$Ct values for each miRNA were split into high and low expression groups based on the corresponding survival data. The `surv_cutpoint()` and `surv_categorize()` functions, available in the `survminer` package in R, were

used to establish the optimal cut-off point for each miRNA and categorize it as high or low expression accordingly [38]. The default `minprop` value of 0.2 was used in `surv_cutpoint()` which indicates a minimum proportion of 0.2, or 20%, of samples must be allocated to either expression group. Next, using `ggsurvplot()`, Kaplan-Meier curves were generated from each miRNA. The log-rank test was used to calculate p-values where a value < 0.05 was considered statistically significant. This was repeated for overall survival, 1-year survival, and disease-free interval (DFI) for all 3 populations. One sample from the CCOGC population was removed from the DFI analysis due to an incomplete medical record.

To assess the diagnostic potential of multiple miRNAs, a decision tree was made to classify samples as either healthy or cases with OSA using the control and OSA samples, respectively. Due to their consistent treatment protocols, both OVC sample groups were used together as a training set. The samples from CCOGC were used as a test set. Additionally, 20% of the control samples (4/21) were withheld to be used in the test set with the remaining 80% being used for training. Using the `rpart` package, decision trees were grown unrestricted allowing all miRNAs to be considered as variables. To avoid overfitting the tree was pruned by selecting the complexity parameter which corresponds with the minimum cross validation error rate. The final tree was used to make class predictions on the unseen test data. Using the `pROC` package, receiver operating characteristic (ROC) curves were generated for training and test data. Accuracy, precision, and recall metrics were calculated with OSA considered to be the "positive class".

MiRNAs operate as a network so we sought to investigate whether a prognostic model containing multiple miRNAs may provide better survival differentiation than a single miRNA. Out of caution, only miRNAs determined to not be associated with hemolysis were considered for these models. All samples from the 3 populations were used together to avoid overfitting. Survival trees were grown and pruned using `rpart()`, as described previously. Trees were made for the same survival metrics as listed above as well as 6-month and 2-year survival. For each survival metric, samples were assigned a group based on the corresponding tree. Kaplan-Meier curves were used to assess differences in survival between the groups. Log-rank tests were performed using `survdiff()` to calculate p-values for each pairwise group comparison. To account for multiple comparisons, the Bonferroni correction was applied. A p-value < 0.05 was considered statistically significant.

Finally, multivariate models were created using both miRNA expression and patient demographic information for the OVC populations only (n = 46). First, univariate Cox regression was used to calculate hazard ratios (HRs) and assess the significance of the patient variables independently. Categorical variables were assessed allowing each category to serve as the reference population. That is, for a variable with n categories there were n HRs calculated, except for categories with only a single sample where a meaningful p-value and HR could not be calculated. Using the same methods as described for the multi-miRNA models, survival trees were grown and pruned using both OVC1 and OVC2 data together. CCOCG was excluded from these models due to inconsistent or missing patient information. Additionally, 2 samples from OVC1 were excluded due to missing ALP activity values. Kaplan-Meier curves were made for significant models, and p-values calculated using the log-rank test for every pairwise group comparison. The Bonferroni multiple comparisons correction was applied and a p-value < 0.05 was considered statistically significant.

## Results

### Group demographic and survival parameters

The breeds represented in all three OSA populations and controls are detailed in S3 Table. For the OSA populations, breeds representing more than one case in at least one population

included mixed breed, Saint Bernard, Golden Retriever, Great Dane, Greyhound, Boxer, and Rottweiler. The distal radius (n = 15) was the most common location in OVC1 and the proximal humerus (n = 3), distal radius (n = 3), and proximal tibia (n = 3) were the most common locations in OVC2. Only the affected bone and not the specific location within the bone was available for the CCOGC patients. In this population, the femur (n = 5) was the most common site of the primary tumour. Overall, in the three OSA populations, there were 29 neutered males, 28 spayed females, 2 intact females and 2 intact males. The median age at amputation was 8.48 years, 8.28 years, and 7 years for OVC1, OVC2, and CCOGC, respectively with the overall age range of 2.66–12.44 years. The median weight closest to the time of amputation was 38.5 kg, 33.2 kg, and 33.4 kg for OVC1, OVC2, and CCOGC, respectively with the overall weight range of 21.0–69.0 kg. The CCOGC samples were treated at four different institutions (Ohio State University (n = 6), University of Wisconsin (n = 4), University of California Davis (n = 2), and Tufts University (n = 1)). All control samples were obtained from OVC and had a median age of 4.58 years (range: 1.22–10.00 years) and median weight at sample collection of 35.1 kg (range: 19.5–59.0 kg), although weight was not available for four dogs which were still considered medium to large based on their known breed.

Complete survival information was available for 57/61 dogs (S4 Table). Three dogs only had a last known date alive, and one dog did not have a reliable history to determine DFI from the CCOGC group. For OVC1, 28/35 (80%) dogs died or were euthanized related to OSA with a median overall survival time of 277 days and a median DFI of 260 days. For OVC2, 11/13 (84.6%) dogs died or were euthanized related to OSA with a median overall survival time of 344 days and a median DFI of 244 days. For CCOGC, 11/13 (84.6%) dogs died or were euthanized related to OSA with a median overall survival time of 754 days and a median DFI of 610 days (12 dogs considered). There were no statistically significant differences between the groups for the median overall survival or median DFI (by one-way ANOVA). For 1-year survival the percent alive in each group varied from 38.5% to 61.5%.

## Differential expression of miRNAs in OSA plasma samples compared to healthy control plasma samples

In the OVC1 population, there were 13 miRNAs and 1 miRNAs with significantly increased and decreased expression, respectively, compared to healthy control dogs (S5 Table). The median fold-difference of the miRNAs with increased expression varied from 1.75 to 6.48 and the miRNA with a decreased expression had a median fold-difference of -1.88. In the OVC2 population, there were 2 miRNAs and 21 miRNAs with significantly increased and decreased expression, respectively compared to healthy control dogs (S5 Table). The median fold-difference of the miRNAs with increased expression was 1.82 and 1.96 and the miRNAs with decreased expression had a median fold-difference that varied from -1.48 to -18.66. In the CCOGC population, 14 miRNAs had a significant increase in expression compared to healthy control dogs, with a median fold-difference varying from 1.98 to 9.80; there were no statistically significant miRNAs with decreased expression in this population (S5 Table). Only miR-214-3p had a significant increase in expression across all three groups compared to controls (Fig 1 and Table 1). The median fold-difference was 5.16, 1.96, and 8.87 for the OVC1, OVC2, and CCOGC populations, respectively. Nine miRNAs had a significant increase in expression in both the OVC1 and CCOGC populations (Table 1).

## MiRNA associations with hemolysis in plasma

It is well established that miR-16-5p, -92a-3p, and -451a are affected by hemolysis so they served as a reference to assess the remaining miRNAs [28–30, 39, 40]. For these 3 miRNAs, the

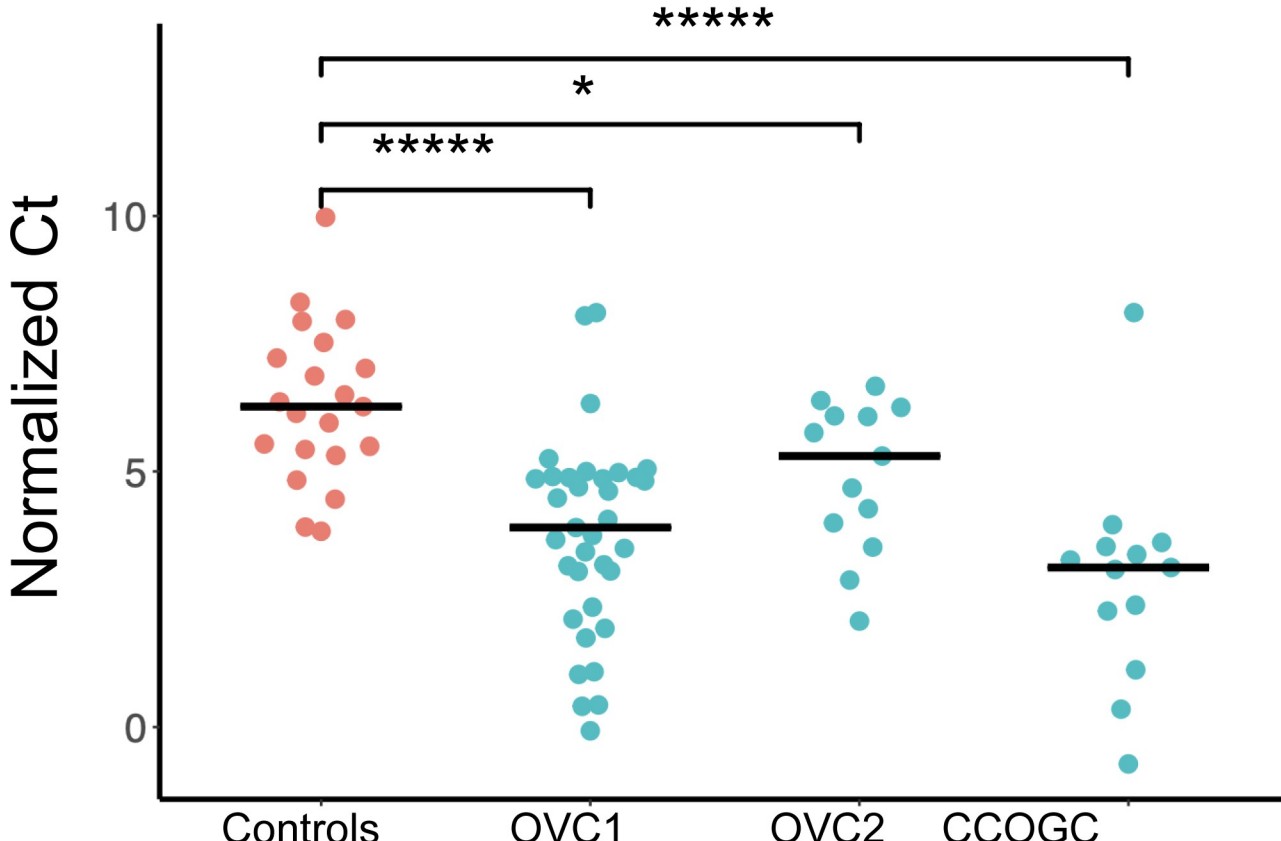

**Fig 1. MiR-214-3p expression across all populations.** The median fold-difference is statistically significant for all three groups compared to controls. The median normalized Ct value is represented by the bar. * p-value < 0.05; ***** p-value < 0.00001.

**Table 1. MiRNAs differentially expressed between OSA populations and healthy controls.**

| | OVC1 | | OVC2 | | CCOGC | |
|---|---|---|---|---|---|---|
| miRNA | Median fold-difference | FDR adjusted p-value | Median fold-difference | FDR adjusted p-value | Median fold-difference | FDR adjusted p-value |
| miR-214-3p | 5.16 | 6.13E-05 | 1.96 | 4.17E-02 | 8.87 | 1.61E-04 |
| miR-133b | 6.48 | 2.06E-04 | | | 9.38 | 6.11E-04 |
| miR-145-5p | 3.77 | 2.06E-04 | | | 3.85 | 5.81E-04 |
| miR-133-3p | 4.24 | 4.52E-04 | | | 9.47 | 4.25E-04 |
| miR-133c | 5.30 | 1.06E-03 | | | 9.80 | 1.14E-04 |
| miR-378a-3p | 2.16 | 8.60E-03 | | | 2.07 | 5.81E-04 |
| miR-30a | 2.86 | 8.60E-03 | | | 3.66 | 1.95E-03 |
| miR-143-3p | 2.49 | 2.50E-02 | | | 5.50 | 6.11E-04 |
| miR-125a | 1.75 | 4.17E-02 | | | 3.46 | 7.77E-04 |

MiRNAs with a statistically significant median fold-difference similarly dysregulated in at least two populations. Only miR-214-3p was statistically significantly and similarly dysregulated in all three populations.

$R^2$ values for a linear trend between expression (Ct value) and 414 nm absorbance ranged from 0.35–0.41. Samples were subsequently separated into non-hemolyzed and hemolyzed groups based on a cut-off (to be considered hemolyzed) 414 nm value of > 0.2 and a 414/375 ratio > 1.4 and evaluated for expression differences for each miRNA. When a Mann-Whitney U test was performed between these groups for miR-16-5p, 92a-3p, and -451a, the p-values were all < 0.005. Based on these values a conservative estimate of association was established where an $R^2$ value > 0.3 and a p-value < 0.01 indicated the expression of a miRNA was highly likely to be influenced by hemolysis (S6 Table). This encompassed 12 of the miRNAs evaluated in the plasma samples. Conversely, an $R^2$ value < 0.2 and a p-value > 0.05 indicated a poor relationship between hemolysis and its expression (S6 Table), which encompassed 21 of the miRNAs evaluated. MiRNAs with values between those used as cut-offs should be treated with caution due to a possible, but undetermined, relationship with hemolysis; this included 14 of the miRNAs evaluated.

## Diagnostic potential of multi-miRNA signatures in canine OSA plasma

In the training set composed of OVC1 and OVC2 plasma samples and 80% of the control samples, using pruned decision trees we found that miR-214-3p and miR-221 in combination was the best at classifying OSA patient plasma samples versus healthy control dogs (area under the curve = 0.872; Fig 2). This combination of miRNAs resulted in the correct classification (accuracy) of 92.3% (60/65) of samples. Specifically, 97.9% (47/48) of the OSA patients were detected (recall) with a positive predictive value (precision) of 92.2%. Only one OSA patient was not captured with this model, however four control samples were classified as positive. This decision tree model was tested on the CCOGC population and the remaining 20% of the control samples. The area under the curve was 0.721 and this combination of miRNAs provided the correct classification of 82.4% (14/17) of samples. Within the CCOGC population, 92.3% (12/13) of the OSA patients were correctly identified (recall) with a positive predictive value (precision) of 85.7%.

## Single miRNAs and clinical features associated with clinical outcome parameters

MiR-214-3p was the only statistically significant miRNA to similarly stratify all three populations into high and low expressing groups for DFI and overall survival (Fig 3; S7 and S8 Tables). High expression of miR-214-3p in plasma was predictive of a shorter median survival time. OVC1 had 21 miRNAs that predicted DFI, of which it shared 7 with OVC2 and 3 with CCOGC (S6 Table). OVC2 had 13 miRNAs and CCOGC had 8 miRNAs which predicted DFI and these two populations did not share any miRNAs which similarly stratified the populations. OVC1 had 21 miRNAs that predicted overall survival, of which it shared 7 with OVC2 and 3 with CCOGC (S8 Table). OVC2 had 16 miRNAs and CCOGC had 9 miRNAs which predicted overall survival. These two populations did not share any miRNAs which similarly stratified the populations. OVC1 had 22 miRNAs that predicted 1-year survival, of which it shared 7 with OVC2 and two with CCOGC (S9 Table). MiR-214-3p was predictive of 1-year survival for the OVC1 and CCOGC populations only, again with high expression predicting shorter survival. OVC2 had 12 miRNAs and CCOGC had 10 miRNAs which predicted 1-year survival and these two populations did not share any miRNAs which similarly predicted 1-year survival. Overall, OVC1 and OVC2 shared the most single miRNAs associated with predicting clinical outcome parameters.

Univariate analyses of multiple clinical parameters were possible for the OVC populations only. The clinical information for the CCOCG population was frequently incomplete or did

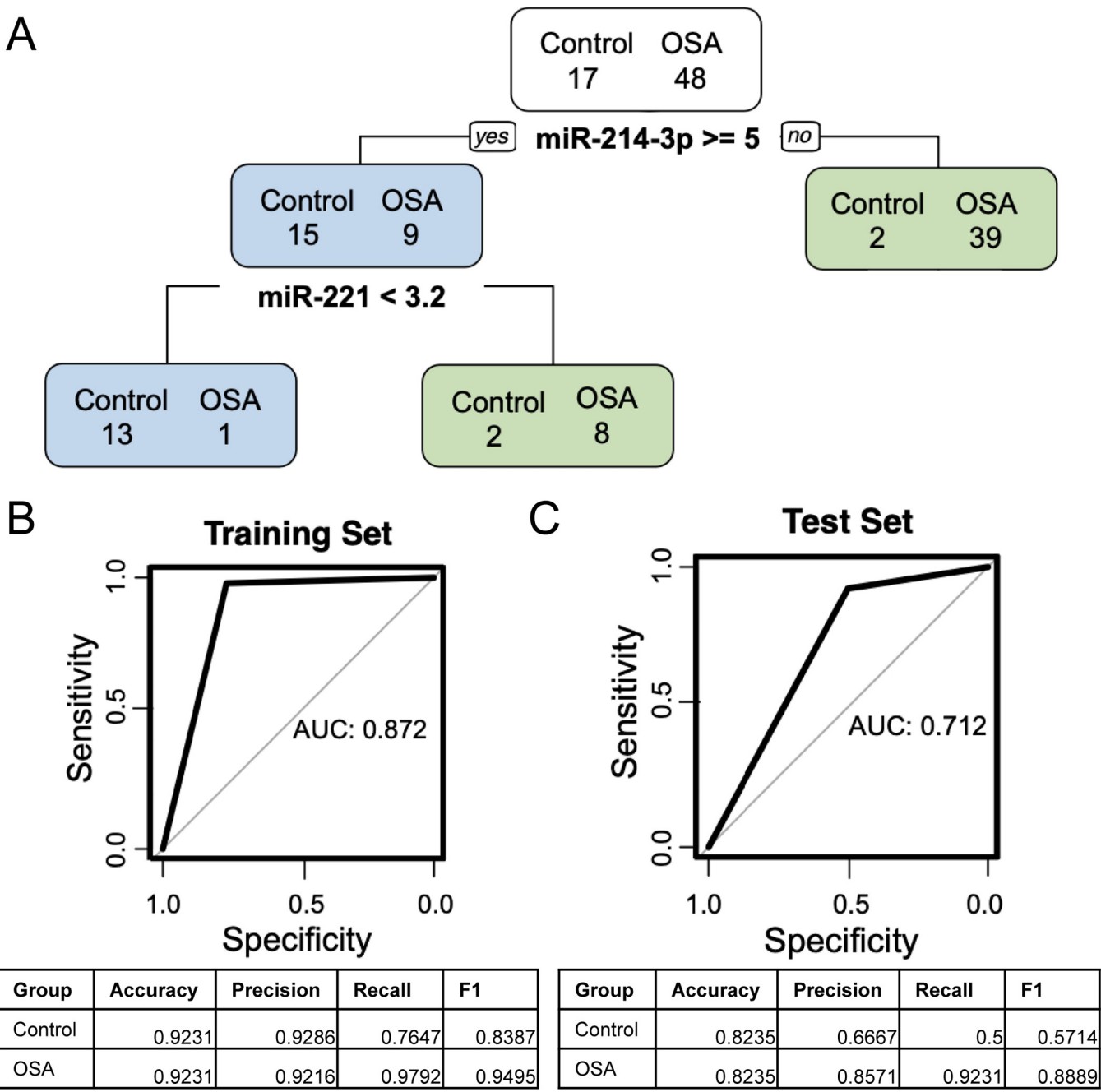

**Fig 2. Pruned multiple miRNA osteosarcoma (OSA) diagnostic model.** (A) Pruned decision tree of the training set (n = 48 OSA and n = 17 control samples) using miR-214-3p and miR-221 (normalized Ct value cut-offs) to classify OSA and control samples. Green boxes represent the model's classification as an OSA sample and blue boxes represent the model's classification as a control sample. (B) Area under the curve graph for the training set with a value of 0.872. (C) Area under the curve graph for the test set (n = 13 OSA and n = 4 control) with a value of 0.712.

not include all parameters evaluated. The clinical parameters evaluated included age, sex (male neutered, male intact, female spayed, female intact), weight, ALP activity (above, within, or below the reference interval), and tumour location (S10 Table). By univariate analysis, tumours located in the humerus and radius had a statistically significant HR for DFI only, with humeral tumours having an increased risk (2.8 (95% confidence interval (CI): 1.2–6.3)) and radial tumours having a decreased risk (0.3 (95% CI: 0.11–0.78)).

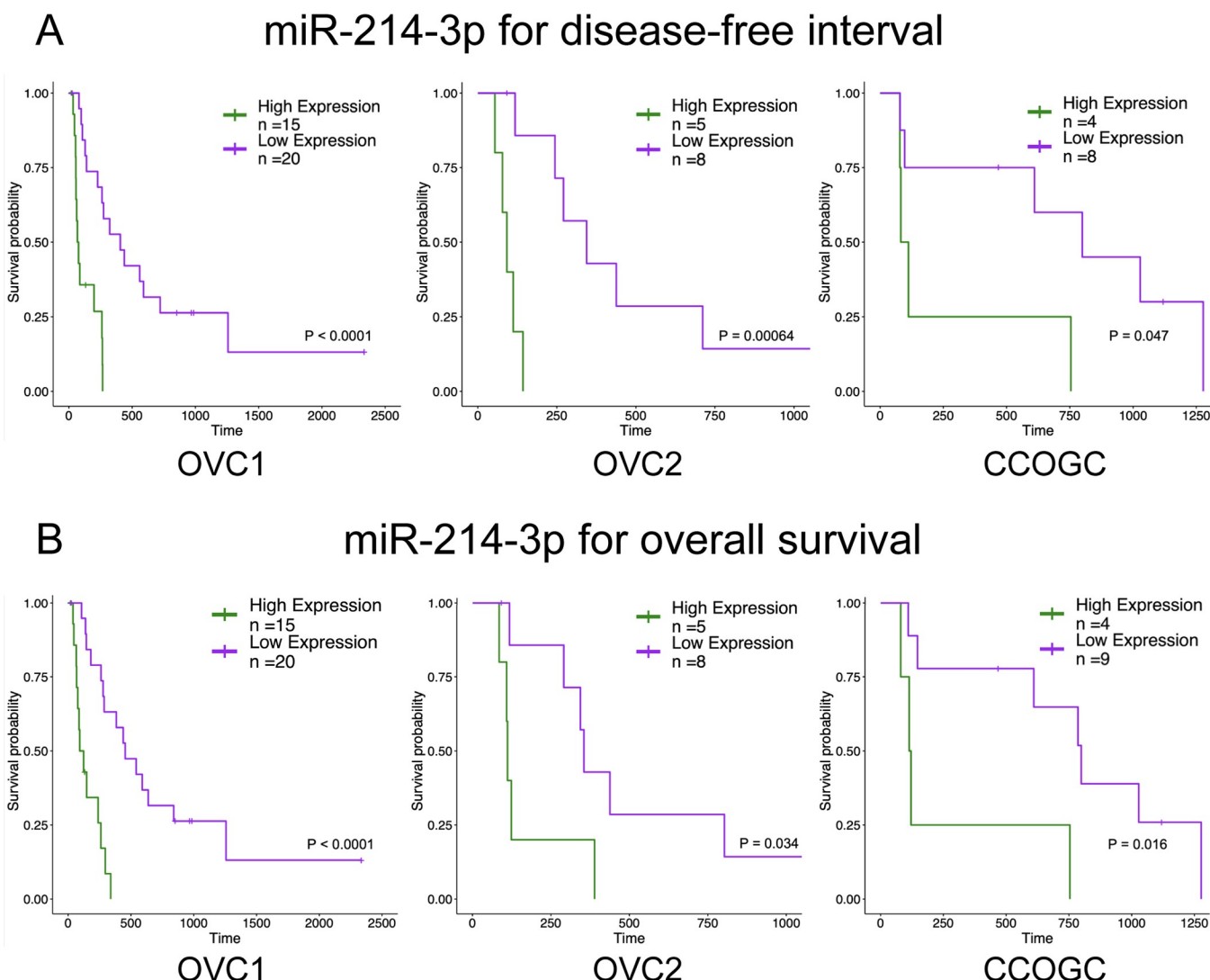

**Fig 3. High expression of miR-214-3p predicts a shorter disease-free interval and overall survival in all three populations.** (A) Kaplan-Meier survival curve for miR-214-3p expression associated with disease-free interval for the OVC1, OVC2, and CCOGC populations. Time in days. (B) Kaplan-Meier survival curve for miR-214-3p expression associated with overall survival for the OVC1, OVC2, and CCOGC populations. Time in days.

## Multiple miRNAs are the best predictors of clinical outcome parameters

As explained above, miRNAs likely associated with hemolysis were excluded from these models to provide a conservative determination of potentially useful miRNAs. All populations were included in the analyses of multiple miRNAs as clinical outcome predictors. Significant models were discovered for overall survival (Fig 4), DFI (Fig 5), 6–month survival, 1-year survival, and 2-year survival (S1 Fig). MiR-214-3p was a significant contributor to all the models, with high miR-214-3p expression always serving as a predictor of shorter survival/increased risk of an event (Figs 4 and 5 and S1 Fig). MiR-214-3p$^{low}$/miR-1307-3p$^{high}$/miR-28-3p$^{low}$ provides the greatest survival probability for overall survival (longest survival; HR: 0.5; Fig 4A) and is significantly different from the other miRNA groups (Fig 4C). MiR-214-3p$^{high}$ is alone an indicator of short overall survival (HR: 2.9), as well as miR-214-3p$^{low}$/miR-1307-3p$^{low}$ (HR: 1.7) and miR-214-3p$^{low}$/miR-1307-3p$^{high}$/miR-28-3p$^{high}$ (HR: 1.7) (Fig 4A and 4B). These

# Multiple miRNA model for overall survival

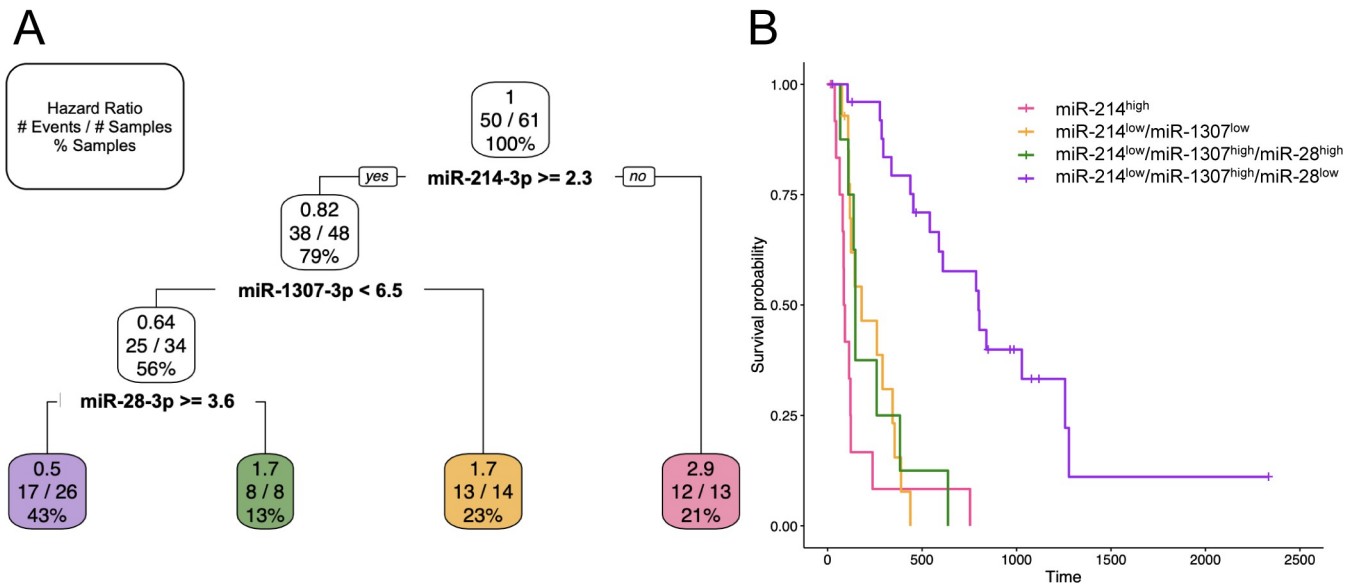

**Fig 4. Multiple miRNA model using miR-214-3p, miR-1307-3p, and miR-28-3p to predict overall survival.** (A) Pruned decision tree for overall survival classified samples into four groups with different hazard ratios (HRs). Each step (normalized Ct values used as cut-offs) includes the corresponding HR, number of events defined by death due to osteosarcoma out of the total number of samples, and percentage samples in each group. (B) Kaplan-Meier survival curve for each of the groups. (C) Median survival time (days) for each group and the group comparisons including HR, 95% confidence interval in parentheses, and corresponding p-value.

three groups are not significantly different from each other (Fig 4C). MiR-214-3p$^{high}$ alone is also an indicator of short DFI (HR: 2.6) (Fig 5A). Those with low miR-214-3p can be further divided into three groups. MiR-214-3p$^{low}$/miR-222-3$^{high}$ has a HR of DFI of 0.37 (Fig 5A) and is significantly different from miR-214-3p$^{high}$ (Fig 5B, 5C). The second group, miR-214-3p$^{low}$/miR-222-3p$^{low}$/miR-93b-3p$^{high}$, has a HR of 0.71 (Fig 5A) and is only significantly different from miR-214-3p$^{high}$ (Fig 5B, 5C). The final group, miR-214-3p$^{low}$/miR-222-3p$^{low}$/miR-93b-3p$^{low}$, is similar to miR-214-3p$^{high}$, and significantly different from the other two low expressing miR-214-3p groups (Fig 5B, 5C).

Further decision trees were investigated for 6-month, 1-year, and 2-year survival (S1 Fig). Cases with miR-214-3p$^{low}$/miR-221$^{high}$/miR-23a$^{low}$ did not experience an event (death due to OSA; HR: 0.074) during the first 6 months after diagnosis (S1A Fig). This miRNA group was significantly different from the three other miRNA groups which were not significantly different from each other (HR range: 1.2–3.1). For 1-year survival (S1B Fig), despite consideration

## Multiple miRNA model for disease-free interval

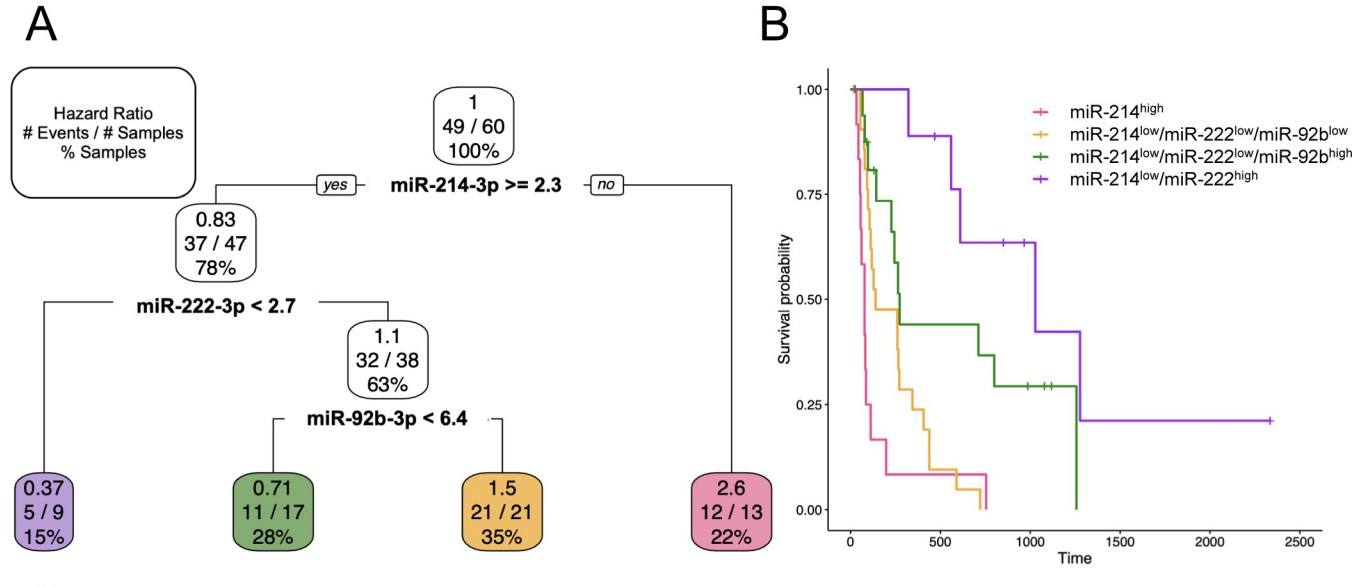

**Fig 5. Multiple miRNA model using miR-214-3p, miR-222-3p, and miR-92b-3p to predict disease-free interval.** (A) Pruned decision tree for disease-free interval classified samples into 4 groups with different hazard ratios (HRs). Each step (normalized Ct values used as cut-offs) includes the corresponding HR, number of events defined by metastatic disease out of the total number of samples, and percentage samples in each group. (B) Kaplan-Meier survival curve for each of the groups. (C) Median survival time (days) for each group and the group comparisons including HR, 95% confidence interval in parentheses, and corresponding p-value.

of multiple miRNAs, miR-214-3p alone was the best predictor of an event occurring, such that high expression of miR-214-3p was predictive of death prior to 1-year (HR: 3.0) and low expression reduced the probability of death by 1-year (HR: 0.74). For 2-year survival (S1C Fig), similar to all other survival parameters evaluated, miR-214-3p^high provided the lowest probability of surviving at least 2-years (HR: 3.5). MiR-214-3p^low/miR-221^low provided a similarly low probability of 2-year survival (HR: 2.3). MiR-214-3p^low/miR-221^high/miR-28-3p^low (HR: 0.49) provided the best probability of achieving a 2-year survival.

Only the OVC populations were used for multiple variable analyses and model parameters included both the clinical variables and miRNA expression. None of the clinical variables (e.g. age, location, ALP activity status) were determined to be valuable in the models and each model was composed solely of miRNAs as the best predictors of clinical outcome (Fig 6). No significant models were found for 6-month survival or 2-year survival. For overall survival (Fig 6A) and 1-year survival (Fig 6C), high expression of miR-214-3p remained the best predictor of a shorter survival (HR: 3.4 for both), along with miR-214-3p^low/miR-221^low (HR: 2.2 for

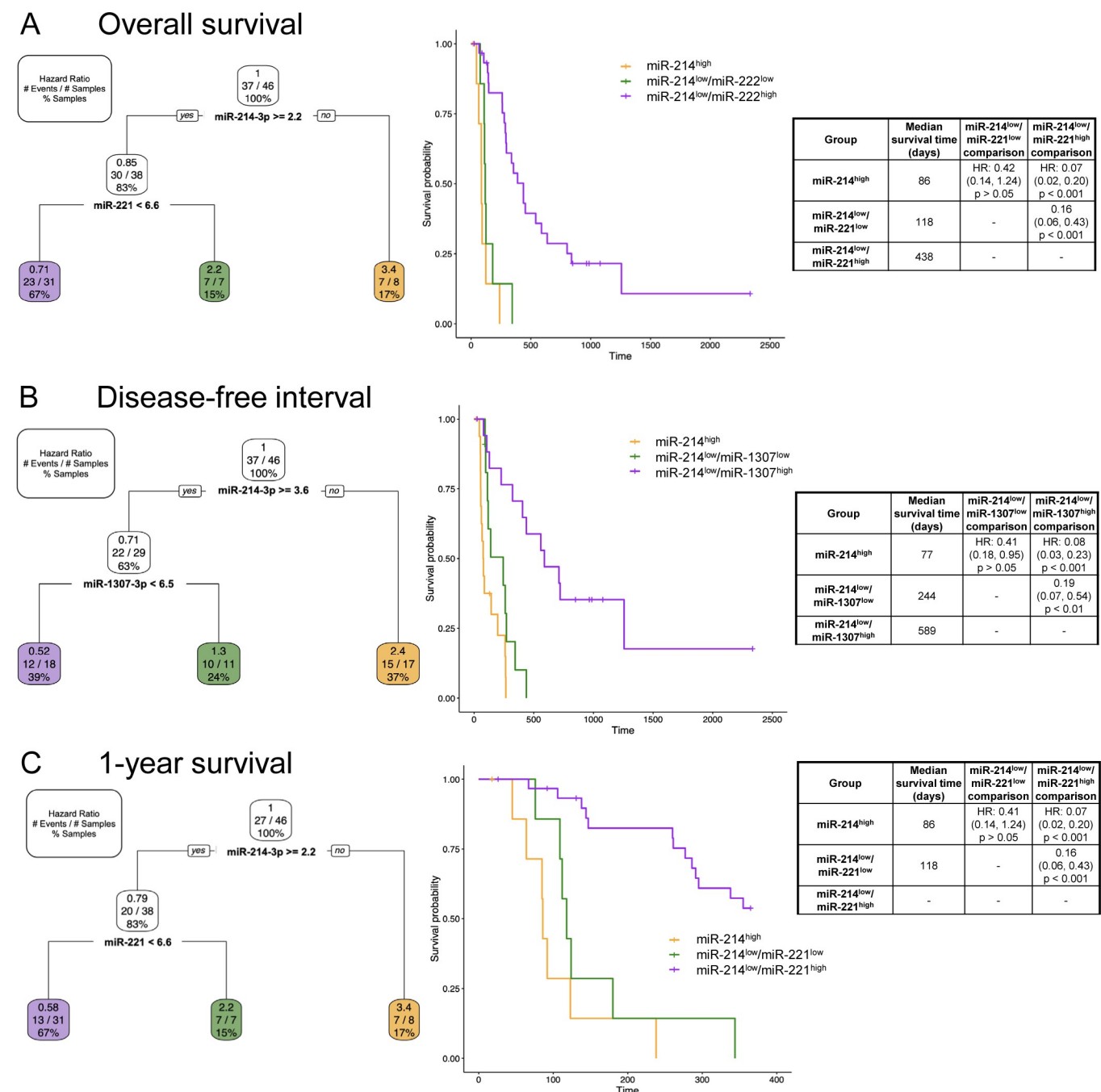

**Fig 6. Multiple variable models for overall survival, disease-free interval, and 1-year survival which only selected miRNAs.** Pruned decision tree. Each step (normalized Ct values used as cut-offs) includes the corresponding hazard ratio (HR), number of events (dependent on clinical outcome parameter) out of the total number of samples, and percentage samples in each group. Kaplan-Meier survival curve for each of the groups. Median survival time (days) for each group, HR, confident interval in parentheses and corresponding p-value between group comparisons. (A) overall survival. (B) disease-free interval. (C) 1-year survival.

both). These were significantly different from miR-214-3p[low]/miR-221[high] dogs which had a hazard ratio of 0.71 for overall survival and 0.58 for 1-year survival. Similarly, for DFI (Fig 6B), high expression of miR-214-3p was a negative prognostic indicator with a HR of 2.4. MiR-214-

$3p^{low}$/miR-1307$^{low}$ dogs also had an increased risk (HR: 1.3). Whereas miR-214-3p$^{low}$/miR-1307$^{high}$ dogs had a better clinical outcome (HR: 0.52).

## Discussion

Multiple miRNAs were dysregulated in all three OSA populations investigation. MiR-214-3p was the only miRNA with increased expression in OSA compared to healthy controls in all three of the populations investigated. Based on our hemolysis evaluation, miR-214-3p was considered not to be associated with hemolysis. This agrees with multiple studies that support the diagnostic potential of circulating miRNAs in cancer, including OSA in humans, mice, and dogs [41–43]. Multiple miRNAs are dysregulated in the blood of patients with OSA [21, 41, 43]. Heishima et al. (2017) reported one of the initial studies that demonstrated differential expression of miRNAs in canine OSA plasma [21]. Their study demonstrated that miR-126 and miR-214 could differentiate between multiple different tumour types and healthy control samples. MiR-214 had particularly high levels in patients with OSA. Similarly, miR-214 is upregulated in the plasma of mice and humans with OSA compared to healthy controls [43].

Our results along with the literature support the diagnostic potential of circulating miRNAs in OSA patients, particularly that of miR-214-3p. The model using miR-214-3p and miR-221, was able to correctly classify 47/48 OSA samples in the training set (OVC populations) and 12/13 of the OSA samples in the test set (CCOGC population). This demonstrates superior recall. A recent meta-analysis evaluating studies investigating circulating miRNAs in humans determined that overall circulating miRNAs have a high diagnostic accuracy and that multi-miRNA models out-performed single miRNAs in predicting OSA [41].

Canine patients with OSA and that do not have radiographic evidence of pulmonary metastases at the time of diagnosis, demonstrate variable outcomes despite standard-of-care therapy [8]. The patients in this study had DFIs ranging from less than two months to over two years without clinically detectable disease. This study found that miR-214-3p expression predicted DFI and overall survival in all three cohorts. Heishima et al. (2019) had similar findings and in both studies, high circulating expression of miR-214-3p indicated a shorter median DFI or overall survival time [22]. A few studies in dogs report circulating miRNAs as predictors of prognosis but do not include cohorts from other institutions [22, 23]. The results in this current study also found numerous single miRNAs that could predict clinical outcome in one group or another, however no others, aside from miR-214-3p, remained statistically significant across all three populations investigated.

All three populations were used to investigate multi-miRNA models to predict clinical outcome. In all models, high expression of miR-214-3p alone was an excellent predictor of a poor prognosis and was the first step of the decision tree. Multi-miRNA profiles determine if a single or multiple miRNAs are better at predicting clinical outcome parameters. Dailey et al. (2021) previously found a two miRNA model in serum (miR-23a and miR-30c) to best predict whether DFI was greater or less than 227 days [23]. Unfortunately, miR-30c was not included in this study and could not be tested on these cohorts. Interestingly, when 6-month survival was examined in this study, similar to Dailey et al. (2021), low expression of miR-23a did provide a survival advantage compared to the group with high expression of miR-23a [23]. In all models, except 1-year survival, additional miRNAs were included to better predict survival than miR-214-3p alone. In the multi-miRNA models, additional miRNAs were able to determine a subset of patients with low expression of miR-214-3p that had a similarly poor prognosis. These additional miRNAs were not consistent across each of the models examined and each may play a different role in the overall biology of this disease.

In multi-variable analyses of our population, only miRNAs were needed to best predict clinical outcome parameters and provided superior predictions compared to any of the clinical patient parameters. To further test the use of multi-variable models, we combined our OVC populations into one group. This combined group of patients all received standard-of-care treatment and clinical characteristics were recorded. These included sex, age, weight, location, and serum ALP activity status. Histopathologic grading schemes have been previously shown not to predict clinical outcome in OSA patients receiving standard-of-care therapy and so were not included as a variable in this study [13]. Only by univariate analysis we found that the tumour location, which predicts clinical outcome in other studies, impacted the hazard ratio [14, 15]. The humerus had an increased hazard ratio and those in the radius had a decreased hazard ratio mirroring findings in other studies. In contrast to only using miRNAs in this study, Heishima et al. (2019) previously determined that the combination of circulating miR-214, miR-126, and ALP activity was more accurate at predicting DFI and overall survival, specifically compared to ALP activity alone [22]. Their study still demonstrates the importance of miRNAs in clinical outcome prediction.

MiR-214-3p expression was not influenced by hemolysis. This is similar to findings reported by Heishima et al. (2019) [21]. This strengthens its utility as a diagnostic and prognostic marker for OSA. MiRNAs are present in erythrocytes and hemolysis influences the miRNA profile of the serum or plasma [28, 29]. Visual inspection is an insensitive and subjective method to determine the degree of hemolysis, while spectrophotometry, with the absorbance measured at 414 nm, is a sensitive and objective method [30, 31]. Lipemia can artificially elevate 414 nm absorbance readings and can be corrected for by evaluating absorbance at 375 nm [32]. Several miRNAs were likely altered by hemolysis, such as miR-92a-3p and miR-505-5p. These miRNAs should therefore be avoided when evaluating patient samples for differential expression, unless the sample are not hemolyzed. Multiple miRNAs require further investigation to confidently determine if they are related to hemolysis. Further investigation is required to find a method to better account for hemolysis in serum and plasma samples to make hemolysis-related miRNAs useful. These miRNAs may be strong predictors of prognosis or helpful for diagnosis, but current methods make them unreliable.

We found very few miRNAs being dysregulated across all three populations. A recent meta-analysis evaluating studies of circulating miRNAs in human OSA patients demonstrated a large variation in miRNAs shown to be dysregulated, highlighting the need for studies that use validation populations [41]. Our populations had differences in both time and location. Frequently, miRNAs were either similarly differentially expressed or provided prognostic prediction within two populations, such as in the OVC1 and OVC2 groups or the OVC1 and CCOGC groups. The OVC2 group had a large number of downregulated miRNAs compared to the control population. This was the oldest collected OVC population and may suggest a loss of miRNA integrity with long-term storage. However, some of the CCOGC samples were collected 2–3 years prior to our OVC2 population samples and did not reflect this similar widespread downregulation of miRNAs. Multiple studies have also demonstrated relatively long-term stability but these studies have only investigated a small number of miRNAs [25, 44]. One study also demonstrated miRNA-dependent degradation in samples stored for 5–14 years [25]. Finally, although miR-214-3p was upregulated in all three populations compared to controls, OVC2 had the smallest median fold-difference, further suggesting some degree of miRNA degradation in this population of samples. Storage time alone does not seem to be the sole contributor and could reflect other differences in handling prior to sample processing. Prospective studies are less likely to have this issue. This highlights one of the limitations of retrospective analyses of miRNAs.

RT-qPCR is an accessible method for investigation of miRNAs, but it is reliant upon selection of miRNAs to be studied. This study investigated a large number of miRNAs by RT-qPCR compared to other canine OSA plasma studies but may still be lacking targets which could be diagnostically and prognostically valuable [21, 22]. Normalization remains a challenge for interpretation of RT-qPCR results and so validation of these results with a second method, such as digital droplet PCR would be useful. Additionally, as these patients all received standard-of-care therapy, the results are only applicable to a population with a diagnosis of OSA without clinically detectable metastases receiving amputation and cytotoxic chemotherapy.

## Conclusion

MiR-214-3p levels in plasma were a diagnostic and prognostic marker for canine patients with appendicular OSA. This miRNA was also a crucial determinant in all multi-miRNA and multi-variable models. This refined the models for prediction of clinical outcome. Hemolysis is an important consideration in the evaluation of serum and plasma samples. Caution is required in the interpretation of miRNAs in variably hemolyzed samples. MiR-214-3p was not associated with hemolysis, making it a clinically useful and practical plasma biomarker for canine OSA.

## Supporting information

**S1 Table. MiRNA assay names included on the canine miScript miRNome (QIAGEN).** (DOCX)

**S2 Table. MiRNA assay names and associated QIAGEN catalog numbers analyzed on the custom arrays.** (DOCX)

**S3 Table. Demographics of the dogs included in each population group.** (DOCX)

**S4 Table. Clinical outcome parameters for each OSA population.** (DOCX)

**S5 Table. MiRNAs differentially expressed between OSA populations and healthy controls.** (DOCX)

**S6 Table. MiRNA associations with hemolysis.** (DOCX)

**S7 Table. MiRNAs associated with disease-free interval in each population.** (DOCX)

**S8 Table. MiRNAs associated with overall survival in each population.** (DOCX)

**S9 Table. MiRNAs associated with 1-year survival in each population.** (DOCX)

**S10 Table. Univariate analyses of clinical characteristics to outcome parameters.** (DOCX)

**S1 Fig. Multiple miRNA models for 6-month, 1-year, and 2-year survival.** (DOCX)

## Acknowledgments

The authors thank the staff of the OVC Veterinary Biobank and CCOGC for providing samples and clinical information, and Dr. Courtney Schott for her collection of clinical information for the OVC patient population. We especially thank the owners and veterinarians for patient samples and for providing care to the patients involved in this study.

## Author Contributions

**Conceptualization:** Michael Edson, R. Darren Wood, Geoffrey A. Wood.

**Data curation:** Michael Edson, Anthony J. Mutsaers, Geoffrey A. Wood.

**Formal analysis:** Latasha Ludwig, Michael Edson, Heather Treleaven, Alicia M. Viloria-Petit, Ayesha Ali, R. Darren Wood, Geoffrey A. Wood.

**Funding acquisition:** Anthony J. Mutsaers, R. Darren Wood, Geoffrey A. Wood.

**Investigation:** Latasha Ludwig, Michael Edson, Heather Treleaven, Alicia M. Viloria-Petit, Anthony J. Mutsaers, Roger Moorehead, Robert A. Foster, Ayesha Ali, R. Darren Wood, Geoffrey A. Wood.

**Methodology:** Latasha Ludwig, Heather Treleaven, Roger Moorehead, Ayesha Ali.

**Project administration:** Geoffrey A. Wood.

**Resources:** Geoffrey A. Wood.

**Software:** Heather Treleaven, Ayesha Ali.

**Supervision:** Anthony J. Mutsaers, Robert A. Foster, R. Darren Wood, Geoffrey A. Wood.

**Writing – original draft:** Latasha Ludwig, Michael Edson, Heather Treleaven, Robert A. Foster, R. Darren Wood, Geoffrey A. Wood.

**Writing – review & editing:** Latasha Ludwig, Michael Edson, Heather Treleaven, Alicia M. Viloria-Petit, Anthony J. Mutsaers, Roger Moorehead, Robert A. Foster, Ayesha Ali, R. Darren Wood, Geoffrey A. Wood.

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
