## [Decision Letter · Decision Letter 0]

28 May 2024

PONE-D-24-13511Plasma microRNA signatures predict prognosis in canine osteosarcoma patients.PLOS ONE

Dear Dr. Wood,

Thank you for submitting your manuscript to PLOS ONE. After careful consideration, we feel that it has merit but does not fully meet PLOS ONE’s publication criteria as it currently stands. Therefore, we invite you to submit a revised version of the manuscript that addresses the points raised during the review process.

**ACADEMIC EDITOR: Please address all the comments by the reviewers for further consideration.**

We look forward to receiving your revised manuscript.

Kind regards,

Nishant Chakravorty

Academic Editor

PLOS ONE

3. We notice that your supplementary tables are included in the manuscript file. Please remove them and upload them with the file type 'Supporting Information'. Please ensure that each Supporting Information file has a legend listed in the manuscript after the references list.

Reviewers' comments:

Reviewer's Responses to Questions

**Comments to the Author**

1. Is the manuscript technically sound, and do the data support the conclusions?

Reviewer #1: Yes

Reviewer #2: Yes

Reviewer #3: Yes

2. Has the statistical analysis been performed appropriately and rigorously? 

Reviewer #1: No

Reviewer #2: Yes

Reviewer #3: Yes

3. Have the authors made all data underlying the findings in their manuscript fully available?

Reviewer #1: No

Reviewer #2: Yes

Reviewer #3: Yes

4. Is the manuscript presented in an intelligible fashion and written in standard English?

Reviewer #1: Yes

Reviewer #2: Yes

Reviewer #3: Yes

5. Review Comments to the Author

Reviewer #1: The authors performed a clinical search of microRNA biomarkers in plasma that could be predictive of OSA diagnosis and prognosis in domestic dogs.

Since my expertise is mostly computational and statistical, although I hold a clinical background, I will focus my review in these aspects throughout the paper, leaving additional minor details for further reviews, in case they might happen in successive rounds. I leave other possibly major issues up to the consideration of additional reviewers with more expertise in the fields I will not touch upon here. Below are my comments:

1.In Table S1, the authors report to have used a panel of 56 probes for their analyses using prior information and a separate trial for expression detection for which there are no shown results in the manuscript, as said in Methods in the “Target miRNA selection” section. Additional information about the miRNA primers used, such as the forward and reverse sequence could be a good addition to the supplementary material, adding clarity to the data exposition.

2.Then, in Table S4, they report a subset of these 56 miRNAs for the differential expression analyses using RT-qPCR data (44). I would like to be able to check the original Ct values and successive normalization steps up until reaching the reported Fold Changes in Table S4 and Table 1. Could the authors provide that? In this way, it could be possible to get access to the “expression” estimates of each miRNA before FC and significance calculation, and therefore better assess the real effect size that a given FC represents. Since high or low FC can greatly vary in their interpretation given the expression levels of the miRNAs in question between groups, adding such information would provide a better source of analysis of the results shown in the manuscript.

3.The authors used a set of microRNAs that include both canine (cfa-) and non-canine (hsa-, bta-, rno-...) probes, which I assume correspond to orthologs of existing expressed canine miRNAs. However, inferring this from the results presented is not straightforward. Could the authors include a supplementary table with the sequence of the mature miRNAs targeted by qPCR, both comparing the canine and non-canine orthologs? To my understanding, these non-canine miRNA primers should target sequences that are conserved among the two species, at least, the domestic dog and the species from which the miRNA primer is derived. It would be really helpful as a cross-check to be able to see that the non-canine miRNAs used are actually equivalent to real existing and expressed canine mature miRNAs circulating in plasma.

4.Presenting the significance of the differential expression the way the authors did could be misleading. First, it would help the reader to get access, both in the main and in the supplementary tables, to the accurate p-value number estimated for each miRNA in each contrast. Simplifying p-values to ranges can obscure differences in significance among different contrasts. A scientific notation of the p-values could also help for a better interpretation. Also, the authors should try to correct their p-values with a multiple testing approach, such as FDR or similar, so as to see whether their estimated p-values survive multiple testing correction thresholding. Maybe the reported cfa-miR-214-3p keeps being the only significant one after FDR correction.

5.Subsetting differential expression analysis tables the way the authors did in the Table 1 could be misleading. I recommend to include all p-values, either significant or not, in scientific notation, removing ranges and providing the real estimate number and adding a column with the multiple testing corrected value (via FDR or similar).

Given these points, I recommend Major revision of the manuscript.

Reviewer #2: Following recommendations are suggested:

1) kindly format the paragraph alignment.

2) arrange the manuscript sections as per journal guidelines.

3) provide the Supplementary data in separate file.

4)it is recommended to include S2,S3,S4 in manuscript, not as supplementary data because this information needed a spotlight.

5) how did you calculate p-value?. Which statistical approach or model is used for statistical analysis?.

6) which significance levels are used as reference?. Kindly mention.

7) kindly arrange table 1 at end of manuscript text,before figures.

Reviewer #3: The paper is well written and explores the use of plasma miRNAs to predict outcomes in dogs with osteosarcoma. It would be useful to include information regarding treatment of the dogs since this may impact their disease progression and survival. Although possibly beyond the scope of this paper, it could be also be informative to discuss the putative functions of the miRNAs that were selected as predictors in the pruned tree models.

6. PLOS authors have the option to publish the peer review history of their article (what does this mean?). If published, this will include your full peer review and any attached files.

Reviewer #1: No

Reviewer #2: **Yes: **Syeda Zahra Abbas

Reviewer #3: No

---

## [Author Response · Author response to Decision Letter 0]

12 Jul 2024

These are all addressed in the uploaded document.

---

## [Decision Letter · Decision Letter 1]

30 Jul 2024

PONE-D-24-13511R1Plasma microRNA signatures predict prognosis in canine osteosarcoma patients.PLOS ONE

Dear Dr. Wood,

Thank you for submitting your manuscript to PLOS ONE. After careful consideration, we feel that it has merit but does not fully meet PLOS ONE’s publication criteria as it currently stands. Therefore, we invite you to submit a revised version of the manuscript that addresses the points raised during the review process.

We look forward to receiving your revised manuscript.

Kind regards,

Nishant Chakravorty

Academic Editor

PLOS ONE

Journal Requirements:

Additional Editor Comments:

Reviewer #1:

Dear Authors,

You have mostly addressed all my previous comments and thereby I will only add three minor additions to be considered for a better clarity and accessibility of your work. In my opinion, they should be implemented before the manuscript is considered acceptable for publication:

1. In line 192, could you please provide a link to the Qiagen miRNA catalog you used that actually directs you to the said catalog? As it is right now it only leads to a generic webpage of the company, which adds no relevant info to the manuscript.

2. I would suggest that authors include Supplementary Tables as Excel tables, as they have done with Table S10, instead of using docx plain files. In this way, it would be easier to get access to the info contained in them.

3. You have added FDR-corrected Pvalues to Table 1, which seems to be a subset of Table S4. In the answer to my initial request no 5 where I asked the FDR values to be included in the main table and the Supplementary Tables, the authors refer that this info has been added to Table S5. I guess they actually meant Table S4, which is the one that includes differential expression analyses results. However, the current Table S4 only includes P-values, and not FDR values, as far as I can see. For this table to be correct and matchin Table 1, you should add the corresponding Pvalues AND the FDR values (both in scientific notation) to all the miRNAs included, which then would lead to the subset shown in the main table in the manuscript.

Given these coments, I consider the manuscript needs Minor Revision.

Best regards.

Reviewers' comments:

Reviewer's Responses to Questions

**Comments to the Author**

1. If the authors have adequately addressed your comments raised in a previous round of review and you feel that this manuscript is now acceptable for publication, you may indicate that here to bypass the “Comments to the Author” section, enter your conflict of interest statement in the “Confidential to Editor” section, and submit your "Accept" recommendation.

Reviewer #1: (No Response)

Reviewer #2: All comments have been addressed

Reviewer #3: All comments have been addressed

2. Is the manuscript technically sound, and do the data support the conclusions?

Reviewer #1: Yes

Reviewer #2: Yes

Reviewer #3: Yes

3. Has the statistical analysis been performed appropriately and rigorously? 

Reviewer #1: Yes

Reviewer #2: Yes

Reviewer #3: Yes

4. Have the authors made all data underlying the findings in their manuscript fully available?

Reviewer #1: Yes

Reviewer #2: Yes

Reviewer #3: Yes

5. Is the manuscript presented in an intelligible fashion and written in standard English?

Reviewer #1: Yes

Reviewer #2: Yes

Reviewer #3: Yes

6. Review Comments to the Author

Reviewer #1: Dear Authors,

You have mostly addressed all my previous comments and thereby I will only add three minor additions to be considered for a better clarity and accessibility of your work. In my opinion, they should be implemented before the manuscript is considered acceptable for publication:

1. In line 192, could you please provide a link to the Qiagen miRNA catalog you used that actually directs you to the said catalog? As it is right now it only leads to a generic webpage of the company, which adds no relevant info to the manuscript.

2. I would suggest that authors include Supplementary Tables as Excel tables, as they have done with Table S10, instead of using docx plain files. In this way, it would be easier to get access to the info contained in them.

3. You have added FDR-corrected Pvalues to Table 1, which seems to be a subset of Table S4. In the answer to my initial request no 5 where I asked the FDR values to be included in the main table and the Supplementary Tables, the authors refer that this info has been added to Table S5. I guess they actually meant Table S4, which is the one that includes differential expression analyses results. However, the current Table S4 only includes P-values, and not FDR values, as far as I can see. For this table to be correct and matchin Table 1, you should add the corresponding Pvalues AND the FDR values (both in scientific notation) to all the miRNAs included, which then would lead to the subset shown in the main table in the manuscript.

Given these coments, I consider the manuscript needs Minor Revision.

Best regards.

Reviewer #2: All the comments have been addressed. I would suggest to accept the manuscript. Only a minor paragraph alignment formatting is suggested as a recorrection. LlPp

Reviewer #3: (No Response)

7. PLOS authors have the option to publish the peer review history of their article (what does this mean?). If published, this will include your full peer review and any attached files.

Reviewer #1: No

Reviewer #2: No

Reviewer #3: No

---

## [Author Response · Author response to Decision Letter 1]

11 Sep 2024

Journal Requirements:

The reference list has been checked for completeness. Two DOIs have been corrected. To our knowledge none of the cited papers have been retracted.

Additional Editor Comments:

Reviewer #1:

Dear Authors,

You have mostly addressed all my previous comments and thereby I will only add three minor additions to be considered for a better clarity and accessibility of your work. In my opinion, they should be implemented before the manuscript is considered acceptable for publication:

1. In line 192, could you please provide a link to the Qiagen miRNA catalog you used that actually directs you to the said catalog? As it is right now it only leads to a generic webpage of the company, which adds no relevant info to the manuscript.

Reply: To access the catalog, you are required to have a free account and log into their system to design a plate to review all of this information. We have now provided the most direct link (line 224); however, this still requires login. This information is freely and publicly available in this fashion and therefore does not benefit being provided again in the manuscript as the probe name and Catalog No. are provided for those interested in replicating or referencing our findings.

2. I would suggest that authors include Supplementary Tables as Excel tables, as they have done with Table S10, instead of using docx plain files. In this way, it would be easier to get access to the info contained in them.

Reply: We have elected to maintain the tables in a Word format for these existing supplementary tables as they are not as large as Table S10.

3. You have added FDR-corrected Pvalues to Table 1, which seems to be a subset of Table S4. In the answer to my initial request no 5 where I asked the FDR values to be included in the main table and the Supplementary Tables, the authors refer that this info has been added to Table S5. I guess they actually meant Table S4, which is the one that includes differential expression analyses results. However, the current Table S4 only includes P-values, and not FDR values, as far as I can see. For this table to be correct and matchin Table 1, you should add the corresponding Pvalues AND the FDR values (both in scientific notation) to all the miRNAs included, which then would lead to the subset shown in the main table in the manuscript.

Reply: Table S5 is the corresponding table to Table 1 in the manuscript. The FDR-corrected values have been added alongside the p-values as requested. These changes have not been tracked as it was extensive re-formatting.

Given these coments, I consider the manuscript needs Minor Revision.

---

## [Editor Report · Decision Letter 2]

13 Sep 2024

Plasma microRNA signatures predict prognosis in canine osteosarcoma patients.

PONE-D-24-13511R2

Dear Dr. Wood,

We’re pleased to inform you that your manuscript has been judged scientifically suitable for publication and will be formally accepted for publication once it meets all outstanding technical requirements.

Kind regards,

Nishant Chakravorty

Academic Editor

PLOS ONE
---

## [Editor Report · Acceptance letter]

5 Oct 2024

PONE-D-24-13511R2 

PLOS ONE

Dear Dr. Wood, 

I'm pleased to inform you that your manuscript has been deemed suitable for publication in PLOS ONE. Congratulations! Your manuscript is now being handed over to our production team.

Kind regards, 

on behalf of

Dr. Nishant Chakravorty 

Academic Editor

PLOS ONE
